# The Absence of FAIM Leads to a Delay in Dark Adaptation and Hampers Arrestin-1 Translocation upon Light Reception in the Retina

**DOI:** 10.3390/cells12030487

**Published:** 2023-02-02

**Authors:** Anna Sirés, Mateo Pazo-González, Joaquín López-Soriano, Ana Méndez, Enrique J. de la Rosa, Pedro de la Villa, Joan X. Comella, Catalina Hernández-Sánchez, Montse Solé

**Affiliations:** 1Cell Signaling and Apoptosis Group, Vall d’Hebron Institute of Research (VHIR), 08035 Barcelona, Spain; 2Centro de Investigación Biomédica en Red sobre Enfermedades Neurodegenerativas (CIBERNED), ISCIII, 28029 Madrid, Spain; 3Departament de Bioquímica i Biologia Molecular, Institut de Neurociències, Facultat de Medicina, Universitat Autònoma de Barcelona (UAB), 08193 Bellaterra, Spain; 4Department of Molecular Biomedicine, Centro de Investigaciones Biológicas Margarita Salas (CSIC), 28040 Madrid, Spain; 5Department of Systems Biology, Facultad de Medicina, Universidad de Alcalá, 28871 Alcalá de Henares, Spain; 6Department of Physiological Sciences, School of Medicine, Campus Universitari de Bellvitge, University of Barcelona, 08907 Barcelona, Spain; 7Institut de Neurociències, Campus Universitari de Bellvitge, University of Barcelona, 08907 Barcelona, Spain; 8Institut d’Investigació Biomèdica de Bellvitge (IDIBELL), Campus Universitari de Bellvitge, University of Barcelona, 08907 Barcelona, Spain; 9Centro de Investigación Biomédica en Red de Diabetes y Enfermedades Metabólicas Asociadas (CIBERDEM), ISCIII, 28029 Madrid, Spain

**Keywords:** retina, rod photoreceptors, Arrestin-1, FAIM, dark adaptation, ubiquitin, light damage, knockout mouse model

## Abstract

The short and long isoforms of FAIM (FAIM-S and FAIM-L) hold important functions in the central nervous system, and their expression levels are specifically enriched in the retina. We previously described that *Faim* knockout (KO) mice present structural and molecular alterations in the retina compatible with a neurodegenerative phenotype. Here, we aimed to study *Faim* KO retinal functions and molecular mechanisms leading to its alterations. Electroretinographic recordings showed that aged *Faim* KO mice present functional loss of rod photoreceptor and ganglion cells. Additionally, we found a significant delay in dark adaptation from early adult ages. This functional deficit is exacerbated by luminic stress, which also caused histopathological alterations. Interestingly, *Faim* KO mice present abnormal Arrestin-1 redistribution upon light reception, and we show that Arrestin-1 is ubiquitinated, a process that is abrogated by either FAIM-S or FAIM-L in vitro. Our results suggest that FAIM assists Arrestin-1 light-dependent translocation by a process that likely involves ubiquitination. In the absence of FAIM, this impairment could be the cause of dark adaptation delay and increased light sensitivity. Multiple retinal diseases are linked to deficits in photoresponse termination, and hence, investigating the role of FAIM could shed light onto the underlying mechanisms of their pathophysiology.

## 1. Introduction

The retina has long been considered as a window to the brain. Several neurodegenerative disorders have also been reported to manifest in the retina, and remarkably, ocular manifestations often precede symptoms in the brain [1,2]. Age is a major risk factor to most blindness and vision loss cases in retinal diseases [3], and in view of our current demographic trends, it is not surprising that retinal pathologies and neurodegeneration are emerging causes of visual impairment in the developing world [4]. Therefore, finding strategies to prevent or delay the onset and progression of retinal pathology and photoreceptor degeneration is a major challenge. Although retinal degeneration has been widely studied, the molecular mechanisms underlying functional alterations and neurodegenerative events involving ubiquitin aggregates and photoreceptor cell death are still relatively unexplored [1,2]. One plausible approach to strengthen our knowledge is to study pro-survival as well as anti-apoptotic modulators that are already expressed and can be potentially modulated to achieve therapeutic effects.

We have previously established how both isoforms of the Fas Apoptotic Inhibitory Molecule, FAIM-S and FAIM-L, act as powerful anti-apoptotic proteins [5,6,7]. While FAIM-S is ubiquitously expressed throughout the organism, FAIM-L (which contains 22 extra N-ter amino acids) is only found in neurons [8]. We previously described that FAIM-L levels are reduced in Alzheimer’s disease patients and mouse models before the onset of neurodegeneration, and how its levels are reduced throughout the Braak stages [7]. Moreover, it was described how FAIM is able to inhibit the formation and dissolve amyloid-beta aggregates in vitro [9], consistent with previous results linking FAIM with neurodegenerative events and Alzheimer’s disease-like models [7,10].

We recently described the potential neuroprotective role of FAIM in photoreceptors, since *Faim* knockout (*Faim* KO) mice for both isoforms present ubiquitin aggregates, chronic gliosis and vascular leakage in the retina. These alterations eventually culminate with a late-onset photoreceptor cell death [11]. Interestingly, FAIM has also been reported to be recruited to ubiquitinated aggregates, and FAIM-deficient cells accumulate ubiquitinated proteins after stress induction [12].

In view of this, it is important to highlight the role of the ubiquitin–proteasome system (UPS) as one of the primary mechanisms by which neuronal proteins are degraded to maintain cellular homeostasis [13]. Although protein degradation is the most well-studied feature of UPS, ubiquitination also serves as an additional post-translational modification involved in many neuronal functions. For instance, UPS is crucial for the synaptic plasticity and self-renewal of neurons [14]. Ubiquitination is also involved in cell signaling regulation by controlling the endocytosis of receptors at the plasma membrane, and ubiquitin modifications can result in alterations of protein–protein interactions or protein localization [15]. 

Protein–protein interactions in phototransduction, the process by which light is converted into electrical signals in the retina, are under a strict temporal and spatial control to sustain the kinetics of the light response, light adaptation mechanisms and photoreceptor cell viability [16]. In addition, it has been reported that ubiquitin participates in regulating the levels of mammalian phototransduction proteins in the retina [17]. For instance, the beta–gamma subunit complex of the G-protein Transducin-α, as well as the G protein-coupled receptor rhodopsin, were shown to be a substrate of ubiquitin-mediated degradation [18,19]. More recent studies have also shown ubiquitination as a regulator of light and dark-dependent translocation of proteins involved in phototransduction in rod photoreceptors. This process has been suggested to contribute to light and dark adaptation, given that signal amplification in the phototransduction cascade seems to be reduced after translocation [20,21]. Nonetheless, the mechanism underlying this process has yet to be fully elucidated. 

Phototransduction and light-dependent protein translocation are fine-tuned processes that need their partakers to act on time and to adjust perfectly in their role. Given that FAIM is involved in ubiquitination events and that FAIM knockout mice present a structural neurodegenerative phenotype in the retina [11], we assessed the potential role of FAIM in retinal function and phototransduction modulation. 

Here, we report that *Faim* KO mice present a delay in dark adaptation from early adult ages. Additionally, aged mice show a decrease in rod-driven and retinal ganglion cell (RGC) responses. We also describe how Arrestin-1 translocation is delayed in *Faim* KO mice, and that Arrestin-1 ubiquitination is completely thwarted in the presence of FAIM in vitro. Moreover, we show that *Faim* KO mice are more susceptible to histopathological alterations and present a differential gene expression landscape after mild light stress induction than WT mice. 

## 2. Methods

### 2.1. Animal Care

All experimental procedures were conducted in compliance with European Union Council directives (2010/63/EU) and with the Spanish legislation for the use of laboratory animals (BOE 34/11370-421, 2013). Protocols were approved by the Ethics Committee for Animal Experimentation of the Vall d’Hebron Research Institute (VHIR) (CEEA 07/23 and 20/15) and the Generalitat de Catalunya (DARP 11769 and 8594).

Homozygous knockout mice for both *Faim* isoforms (*Faim* KO mice), bred in a mixed C57BL6/J and 129 background, were kindly provided by Dr. Lam [22] and crossbred with C57BL/6J OlaHsd wild-type mice (WT) (Envigo, France) (IMSR Cat# JAX:000664) for at least 15 generations before use. Heterozygous mice were used to breed *Faim* KO mice and WT littermates for experimental procedures. Animals were bred and maintained in the animal facility at VHIR. They were housed in groups of a maximum of five animals under standard laboratory conditions at 22 ± 2 °C, a 12 h light/dark cycle (8 a.m. to 8 p.m.), a maximum light intensity of 350 lux and a relative humidity of 50–60%. Animals had access to water and food ad libitum (Teklad Global 18% Protein Rodent diet, Envigo, France). An equal number of females and males per study group was selected, and at least four animals per genotype and condition were used for each experiment.

Genotypes were confirmed by PCR analysis. A tail snip was used for DNA extraction. The tail sample was digested in an alkaline solution with Proteinase K (0.24 Units/sample) (Qiagen, Hilden, Germany) for 120 min at 55 °C. The following primers were used to perform a semiquantitative PCR: FAIM-P1-Fw 5′-GAGACTGAGACAGGAGAAGCC-3′; FAIM-P2-Fw 5′-GCTCTTCAGCAATATC-ACGGG-3′ and FAIM-P3-Rev 5’-GCTCAGGTTAAGTGAAGTGCG-3′.

### 2.2. Electroretinography

Retinal function was assessed by electroretinography (ERG), which measures the electrical activity of the retina in response to a light stimulus. Mice were dark-adapted overnight in a separate dark room, and all experimental procedures were performed under a dim red light (ranging from 6.5 to 13 lx). Right before ERG analyses, mice were anesthetized intraperitoneally with a mixture of ketamine (37 mg/kg) and xylazine (10 mg/kg). The pupils were dilated with tropicamide drops (Colircusí TM, Alcon Cusí S.A., Barcelona, Spain). Mice were placed inside a Faraday box on a heating pad maintained at 37 °C. The ERG response of the right eye was recorded using a golden loop electrode, which was placed over the cornea. A reference electrode was placed on the tongue, and a ground electrode was inserted subcutaneously parallel to the tail. Both corneal surfaces were protected with a drop of non-irritating ionic conductive solution (contact lens solutions containing sodium chloride and 0.5% methylcellulose (Methocel 2%, Omnivision, Santa Clara, CA, USA). 

Full-field retinal stimulation was achieved by a dome-like Ganzfeld stimulator placed in front of the right eye. Two different ERG protocols were used: standard flash stimuli protocol and a double-flash protocol to assess dark adaptation.

ERG recordings were obtained using a device designed by Dr. P. de la Villa. Electrophysiological signals were amplified and band-pass filtered between 0.3 and 1000 Hz by a CP511 AC amplifier (Grass Instruments, Quincy, MA, USA), and subsequently digitized to 10 kHz with a PowerLab 4/35 data acquisition card (AD Instruments Ltd., Oxfordshire, UK). Data was acquired, digitized and stored using LabChart Pro v.7 (AD Instruments Ltd., Oxfordshire, UK) software.

#### 2.2.1. Standard Flash Stimuli Protocol

Standard ERG protocols were adapted from previous publications [23,24], and are based on the standard protocols advised by the International Society for Clinical Electrophysiology of Vision (ISCEV) [25]. For dark-adapted (scotopic) conditions, mice were dark-adapted overnight and stimulated with flashing lights ranging from –5 to 2 log scotopic candlepower (cd·s/m^2^) emitted by a white LED inside a Ganzfeld dome [23,24]. Each range of light intensities allowed the recording of currents arising from specific retinal cell types. In dark-adapted ERG recordings, the positive scotopic threshold response (pSTR) was recorded from −5 to −4 log cd·s/m^2^; rod-mediated responses from −4 to −1.5 log cd·s/m^2^; mixed rod and cone responses from −1.5 to 1.5 log cd·s/m^2^. Oscillatory potentials (OPs) were obtained by applying a 100 Hz high-pass digital filter on 1.5 log cd·s/m^2^ mixed responses on LabChart software. There was a pause of 3 to 150 s between flashes depending on the light intensity to allow photoresponse recovery. For each intensity, electrical responses after 4–60 consecutive light flashes were averaged.

For light-adapted (photopic) ERG recordings, a background light of 30–40 cd/m^2^ was applied for four to five minutes to suppress rod stimulation prior to flashing light stimulation. Light intensities ranging from −0.5 to 1.5 log cd·s/m^2^ were used to obtain cone-mediated electrical responses. Flicker responses were recorded at a 20 Hz frequency and 1.5 log cd·s/m^2^ intensity. Each response was averaged between 40–60 times with 1.2 s interstimulus time. 

The amplitude of the waves of each recorded electroretinography response was measured by LabChart software. pSTR were measured from the baseline to the peak of the following positive deflection after −5 and −4 log cd·s/m^2^ light stimuli. The amplitude of the a-wave, which stems from photoreceptor hyperpolarization, was measured from the basal registry point to the following trough after the light flash. The b-wave amplitude, which stems mainly from bipolar depolarization, was measured from the trough of the a-wave to the peak of the following positive deflection. The amplitude of each OP was considered as the difference between its maximum peak and the immediately preceding negative peak. Flicker activity was obtained by averaging three different ERG responses.

The a-, b-wave and OPs implicit times (or peak times) were measured from the flash midpoint to the trough or peak of the corresponding wave.

An equal number of females and males per study group was selected. At 18 months of age, seven to 14 mice were used per genotype. At three months of age, eight to nine mice were used per genotype.

#### 2.2.2. Double-Flash ERG for Dark Adaptation Assessment

For the assessment of dark adaptation and the recovery time of rod photoreceptors, a double-flash protocol was performed [26,27] with some modifications. A first flash (test) was delivered to desensitize rod photoreceptors. Then, a second flash (probe) was delivered to monitor the recovery of the rod response. The time interval between the two flashes of light increased with each set, ranging from 1 to 150 s. The intensity of the light flashes was 1.5 log cd·s/m^2^, and there was a delay with each subsequent set of double-flash ranging from two to five minutes for adequate rod response recovery. Data are represented as log(a/a_max_*100) or log(b/b_max_*100) to interstimulus time, where “a_max_” and “b_max_” are the a- and b-wave after the test flash, and “a” and “b” are the a- and b-wave after the probe flash. 

### 2.3. Light Exposure Protocol

Three-month-old *Faim* KO and WT mice were dark-adapted overnight and exposed to bright light in a cage with white LED panels (2500 lux) for different time periods 5 or 15 min after their pupils were dilated with a topical drop of 1% tropicamide, a mydriatic drug. For the dark-adapted condition, mice were euthanized after overnight dark adaptation. Eyes were collected in the dark, under a dim infrared light for immunofluorescence analysis. An equal number of females and males per study group was selected, and four mice were used per experimental condition.

### 2.4. Light Damage Protocol

For the mild light damage (LD) protocol, three-month-old *Faim* KO and WT mice were dark-adapted overnight and then exposed to white LED light at 10,000 lux in non-reflective cages for 8 or 24 h after their pupils were dilated with drops of 0.5% tropicamide and 2.5% phenylephrine hydrochloride. During the LD protocol, mice were allowed free access to food and water.

For ERG measurements and consequent sample collection for biochemical or immunohistological analyses, mice were kept in their cages after 8 h LD for one week and dark-adapted overnight before the ERG procedures.

For RNAseq analyses, mice were exposed to 24 h LD and immediately euthanized afterwards. Left-eye retinas were rapidly collected and submerged in 1 mL of RNAlater (Thermofisher, Waltham, MA, USA) and maintained at 4 °C overnight before freezing at −20 °C for a maximum of five days. Right-eye retinas were collected, snapped frozen in liquid nitrogen and stored at −80 °C until further use.

An equal number of females and males per study group was selected. Four to five mice were exposed to 8 h of LD per genotype and three to four mice were exposed to 24 h of LD per genotype. All animals exposed to LD were used for the subsequent experiments.

### 2.5. Tissue Collection and Processing

Mice were deeply anesthetized in an induction chamber with 5% isoflurane (Ecuphar, Sant Cugat del Vallès, Spain) delivered with 21% oxygen and transcardiacally perfused with ice-cold saline (0.9% NaCl). Then, mice were decapitated and eyes were enucleated. The retina was extracted from the left eye and snapped frozen in liquid nitrogen for protein or mRNA extraction, unless stated otherwise. The right eye was fixed in 4% paraformaldehyde for 5 h, washed in PBS, dehydrated in a rising ethanol series and embedded in paraffin blocks. Next, 4-μm-thick sections were cut on a microtome (Microm HM 355 S, ThermoFisher Scientific, Waltham, MA, USA), and three or four sections were placed on one slide. Slides were stored at room temperature until the analysis. Experiments were performed using samples from 3- or 18-month-old mice, and at least four animals per group were used in each experiment.

### 2.6. Immunofluorescence and Quantification

Briefly, slides were heated at 60 °C for 1 h, deparaffinized in xylene and rehydrated in a graded ethanol (EtOH) series. For GFAP and Ubiquitin immunodetection, slides were post-fixed in an ice-cold methanol–acetic acid solution (95% methanol, 5% glacial acetic acid). Antigen retrieval was achieved by immersing the slides in an antigen retrieval solution (0.1 M citrate buffer pH 6.0) and placing them in a pressure cooker at 150 °C for 4 min. For Arrestin-1 and Transducin-α immunodetection, slides were placed in a glass-staining container with an antigen retrieval solution (PBS with 0.1 mg/mL proteinase K (Roche)) at room temperature for two minutes and slides were heated at 70 °C for 6–8 s individually. Slides were placed in a humidity chamber and non-specific binding sites were blocked for 1 h at room temperature with PBS containing 0.2% Triton X-100 and 1% BSA (GFAP and Ubiquitin) or 5% BSA (Arrestin-1 and Transducin-α).

After blocking, sections were incubated with the corresponding primary antibody overnight at 4 °C (for antibody details, see Appendix A). The following day, sections were incubated for 1 h at room temperature in the dark with the fluorescent secondary antibodies (for antibody details, see Appendix A).

Nuclei were stained with Hoechst 33342 (0.05 µg/mL, ab228551, Abcam, Cambridge, UK) at 1:10,000 in PBS for 10 min. Sections were mounted with Prolong™ Gold Antifade Mountant (P36930, Thermofisher Scientific, Waltham, MA, USA). Samples without primary antibodies were used as negative controls. Images were taken at 40× in a Zeiss LSM 700 Confocal microscope. GFAP signal intensity was quantified with Fiji, and ubiquitin cells were individually counted in a blinded manner.

For Arrestin-1 and Transducin-α, signal intensities were measured and quantified using ImageJ software (National Institutes of Health, NIH, Bethesda, MD, USA). Signal was quantified using the Region of Interest (ROI) tool of ImageJ, which was used to outline the whole photoreceptor inner (IS) and outer segments (OS) of the retinal image to calculate the percentage of Arrestin-1 and Transducin-α signal located in the OS.

### 2.7. TUNEL Assay

Slides were deparaffinized and rehydrated as above. They were then permeabilized for 8 min in 0.1% sodium citrate containing 0.1% Triton X-100. Slides were afterwards placed in a plastic jar containing 200 mL of 0.1 M citrate buffer pH 6.0 and irradiated at 750 W in a microwave for 1 min for antigen retrieval, and immediately cooled by adding room temperature ddH_2_O.

Slides were immersed in Tris-HCl 0.1 M pH 7.5 containing 3% BSA and 20% fetal bovine serum for 30 min at room temperature for blocking and then washed with PBS. Slides were incubated for 1 h at 37 °C with a TUNEL (Terminal deoxynucleotidyl transferase dUTP Nick End Labeling) reaction mixture (11767291910 and 11767305001, TUNEL Label Mix and TUNEL Enzyme, Roche), following the manufacturer’s instructions. Slides were counterstained with Hoechst 33342 and mounted in Prolong™ Gold Antifade Mountant. An amount of 40× images were taken with the Zeiss LSM 700 Confocal microscope, and TUNEL-positive cells in three different retinal sections were manually counted in a blind manner using ImageJ software. Three distinct whole retinal sections per mouse were analyzed and averaged, and at least four animals per group were used.

### 2.8. Western Blotting

Retinal proteins were homogenized with a RIPA lysis buffer (150 mM NaCl, 1% IGEPAL^®^ CA-630, 0.5% sodium deoxycholate, 0.1% SDS and 50 mM Tris pH 8.0) containing a 1x cOmplete™ EDTA-free protease inhibitor cocktail (11873580001, Roche, Darmstadt, Germany) and collected discarding the pellet after centrifugation at 9300× *g* for 15 min at 4 °C. Protein concentration was quantified using the Modified Lowry kit (DC protein assay; Bio-Rad, Hercules, CA, USA).

Next, 20 to 40 µg of total protein was resolved by SDS-PAGE and transferred onto PVDF Immobilon-P transfer membranes (Millipore Iberica, Madrid, Spain). Membranes were blocked with Tris-buffered saline with 0.1% TWEEN^®^-20 and 10% non-fat dry milk for 1 h at room temperature and then probed overnight with the appropriate primary antibody (for antibody details, see Appendix A). 

Blots were then incubated with the corresponding peroxidase-conjugated secondary antibodies (for antibody details, see Appendix A) for 1 h at room temperature, developed using the EZ-ECL Enhanced Chemiluminescence Detection Kit for HRP (Biological Industries, Kibbutz Beit Haemek, Israel). Band intensities were quantified with ImageJ software [28]. To control equal loading, housekeeping α-tubulin or Ponceau staining were used to normalize protein expression.

### 2.9. RNA-Sequencing and Analysis

Immediately after LD protocol (24 h at 10,000 lux) or normal light exposure, mice were deeply anesthetized in an induction chamber as stated previously. Then, mice were decapitated and eyes were enucleated, and retinas were rapidly obtained and preserved in RNAlater at 4 °C overnight and frozen the following day. Total RNA was obtained using the RNeasy kit following the manufacturer’s instructions. RNA concentration was determined with the NanoDrop spectrophotometer (ThermoFisher Scientific, Waltham, MA, USA). RNA integrity number (RIN) and quality assessment, library preparation and Illumina sequencing (Agilent Technologies, Santa Clara, CA, USA) were performed in the Genomic Unit of the Centre de Regulació Genòmica (CRG) facility.

A cDNA library for every RNA sample was made using 5–10 μg of RNA to prepare the RNA-seq library using TruSeq RNA Sample Prep Kits (Illumina). First, the RNA was isolated from the sample and traces of contaminating DNA were removed with DNase. The polyA containing mRNA molecules were purified using poly-T oligo-attached magnetic beads. Then, the mRNA were reverse transcribed into double-stranded cDNA, and sequence adapters were added to the ends of the fragments. The cDNA library was made by synthesizing the first-strand cDNA using reverse transcriptase and random primers, and followed by synthesis of the second-strand cDNA using DNA polymerase I and RNase H. After construction of the libraries, paired-end sequencing on the Illumina HiSeq 2000 platform was performed using the nucleotide sequences of the ends of the fragments, from now on called “reads”, and 2 × 100 bp reads were obtained from each sample.

The raw reads (in fastq format) were aligned with an indexed mouse reference genome (GRcm39) with Spliced Transcripts Alignment to a Reference (STAR) [29]. Transcript quantification and differential analysis of count data were performed with the count-based statistical method DESeq2 [30] in RStudio [31]. An exploratory data analysis for quality assessment, as well as to explore the relationship between the samples, was made. Variance stabilizing transformation was performed with rlog and VST, and sample distances to assess similarity were calculated with VST data and plotted with a principal components analysis (PCA).

### 2.10. Cell Culture Conditions

HEK293T cells were cultured in DMEM (Cat# CRL-3216, ATCC), supplemented with 10% of heat-inactivated fetal bovine serum (FBS) (A4766801, ThermoFisher Scientific), 20 U/mL penicillin and 20 ug/mL streptomycin (K4000, Sigma Aldrich, St. Louis, MO, USA).

To subculture cells, the medium was removed, and dissociation was performed in a 1:5 dilution of 0.25% trypsin-EDTA (25200, ThermoFisher Scientific) in PBS. Cells were incubated at 37 °C until they were detached and the enzymatic effect of trypsin-EDTA was blocked with an FBS-containing fresh culture media. 

### 2.11. Cell Transfection and Ubiquitin Assay

Ubiquitinylated proteins were detected using the 6 × His-ubiquitin assay for the mammalian cell protocol of the Tansey laboratory (College of Medicine, University of Florida). Cells were seeded at 2.5 × 10^6^ confluency in a P100 dish and left to attach overnight. The following morning, they were transfected with the desired expression plasmids using polyethylenimine (PEI) in DMEM without FBS for 6 h. The following plasmids were used, according to the conditions specified for the experiments: 6×His-ubiquitin, HA-Arrestin-1 (kindly provided by V. V. Gurevich, Department of Pharmacology, Vanderbilt University), FAIM-L-FLAG or FAIM-S-FLAG. When needed, empty vector pCDNA3 was co-transfected to reach 15 μg of total DNA in each condition. After transfection, the media was removed and cells were maintained with a supplemented culture media for 48 h. To achieve proteasome inhibition, the culture media was replaced with a supplemented medium containing 10 μM of MG132 (Cat# HY-13259, Sigma Aldrich, St. Louis, MO, USA) from a stock solution of 10 mg/mL of MG132 diluted in DMSO (D2650, Sigma-Aldrich) and cells were incubated for 6 h.

Before ubiquitin assay was performed, Anti-His affinity resin was washed in buffer A (6 M guanidine-HCl, 0.1 M Na_2_PO_4_, 10 mM Imidazol, pH 8.0) three times by shaking the tube by hand for 10 s. After centrifugation for one minute at 1000× *g*, the resin was pelleted and resuspended in buffer A in a 1:1 ratio in the final wash.

Then, ubiquitin assay was performed as follows. Cells were collected in PBS pH 7.0 and spun down at 1000× *g* for 5 min. Cell pellets were resuspended in PBS and 20 μL of input was stored. Cells were spun down again at 1000× *g* for five minutes and pellets were resuspended in buffer A. Samples were sonicated and homogenized, and lysates were centrifuged at 16,000× *g* for 15 min at 4 °C. Supernatants were transferred to a new tube and resuspended Anti-His affinity resin was added. 

Cell lysates were orbitally rotated for 4 h at 4 °C at 25 rpm. After a short spin, supernatant was discarded, and resin was washed twice with 800 μL of buffer A, four times in buffer A/T1 (1 volume of buffer A for 3 volumes of buffer T1) and twice in buffer T1 (25 mM Tris-HCl, 20 mM Imidazol pH 6.8). After a short spin, a needle was used to aspirate the supernatant and 37.5 μL of T1 buffer supplemented with 25% Imidazol 1 M was added to the resin. An amount of 12.5 μL of 4x Laemmli buffer (60 mM Tris-HCl pH 6.8, 4 mM EDTA, 10% glycerol, 2% SDS, 100 mM dithiothreitol (DTT) and traces of Bromophenol blue) was added to IP lysates and stored inputs, and boiled at 95 °C for 10 min for elution. Afterwards, lysates were loaded in SDS-PAGE for western blot analysis.

### 2.12. Statistical Analyses

We used a minimum number of mice for experimental analysis based on preliminary data and mice availability. Data sets were plotted and analyzed using GraphPad Prism v8.0.1, and no data was removed prior to analysis. Data was graphed before the statistical test to check whether it met the assumptions of the statistical approach. Statistical comparisons were made using a Two-way ANOVA test followed by Sidak’s post hoc test when needed. Three-way ANOVA was used to assess three factors in electroretinogram analyses: light intensity, LD and genotype. An unpaired two-tailed Student’s *t* test was used for parametric data when assessing only one factor. All *p* values lower than 0.05 were considered statistically significant. The ROUT method (with Q set to 1%) was used to detect outliers.

## 3. Results

### 3.1. Loss of FAIM Leads to Age-Related Rod Photoreceptor and RGC Functional Defect

To assess whether the absence of FAIM in the retina leads to functional deficits, we performed ERG analyses using a wide range of light intensities to evaluate the specific response of the retinal neurons (Figure 1). In previous studies, we observed that structural and molecular alterations compatible with neurodegeneration occurred gradually in *Faim* KO retinas and peaked at 18 months, when photoreceptor cell death was detected. In sight of this, we first analyzed the responses of 18-month-old *Faim* KO mice.

ERG analyses revealed reduced amplitudes of both the mixed a- and b-waves (linked to both rod- and cone-driven responses) in aged *Faim* KO mice in comparison to WT littermates (Figure 1A.i,B), while implicit times seemingly remained unaltered (Figure 1C). However, photopic b-wave amplitudes, which are exclusively driven by cones, were similar between *Faim* KO and WT mice (Figure 1B). Similarly, the flicker waves recorded no alterations in aged *Faim* KO mice (Figure 1B). These results indicate that the impaired photoreceptor response in aged *Faim* KO mice is linked to rod, but not cone, photoreceptors, and suggests that FAIM is important for the maintenance of rod function.

We then checked the b:a ratio in scotopic conditions, which is used to assess electronegative ERG waves and whether the system dysfunction is found post-phototransduction. No differences in the b:a ratio between the *Faim* KO and WT were found (Appendix A) suggesting that alterations may stem from the photoreceptor response.

To detect a clean and unambiguous pSTR, a low light intensity was used (Figure 1A.ii). The pSTR peak amplitude, which is mostly associated to RGC responses, was reduced in *Faim* KO mice in comparison to WT, suggesting that RGC function is also compromised (Figure 1B). Moreover, the OPs analysis (Figure 1A.iii) showed a significant decrease in the amplitude of OP4, OP5 and OP6 in aged *Faim* KO animals, while no differences were observed in the amplitude or implicit times of OP1, OP2 and OP3 (Figure 1D,E). OP4 to OP6 are considered to be linked to amacrine and RGC responses. In addition to reduced amplitude, FAIM KO mice also exhibited increased implicit time for OP4, OP5 and OP6, which could evidence a certain delay in the amacrine and RGC response to the light stimulus. Hence, these results suggest that the RGC function is also compromised by the loss of FAIM, but we cannot rule out the possibility that amacrine cells may also be affected.

Given that the neurodegenerative features that we detected at the histological level in *Faim* KO retinas occur progressively [11], we wondered whether retinal function could already be impaired at younger ages, when *Faim* KO mice already present a gliotic phenotype. Retinal function was examined at three months (Figure 2 and Appendix A), but no alterations in the ERG amplitudes were found for any of the analyzed waves, namely the rod-driven response (mixed a- and b-wave and scotopic b-wave) (Figure 2A–C), the RGC responses (pSTR and OPs) (Figure 2D–G) and cone-driven responses (photopic b-wave and Flicker) (Appendix A). This suggests that the retinal functional alterations found in the rods and RGC of 18-month-old *Faim* KO mice are associated to ageing, which is in accordance with the molecular alterations and photoreceptor cell deaths that we found at this age [11]. Interestingly, we did find a significant increase in the mixed b-wave implicit time in *Faim* KO mice (Figure 2E, right), implying that there is a delay in the rod-driven response at three months of age.

Besides being associated with ageing processes, retinal degeneration can also be caused or influenced by extrinsic environmental conditions such as light damaging conditions [32,33]. Light stress induces the expression of endogenous molecular factors that confer protection to the retina and the eye [34,35]. We previously showed that Endothelin-2 (EDN2) and FGF2 are upregulated in the retinas of *Faim* KO mice without any additional stress [11]. Interestingly, these proteins have been proposed to act as retinal neuroprotective factors after LD exposure [35,36,37]. In sight of this, we decided to study whether *Faim* KO mice show increased sensitivity to LD. To test this, we subjected young adult (three-month-old) mice to mild LD (10,000 lux for 8 h) and their visual function was assessed one week later. However, we did not find any significant decrease in ERG amplitudes between groups or after LD in scotopic, mixed or photopic responses or OPs (Appendix A). 

### 3.2. Faim Depletion Causes a Delayed Dark Adaptation 

A closer look to the ERG recording profiles of young adult animals showed that the decay of electrical activity after light flashes differed between *Faim* KO and WT mice (Figure 3A). Soon after phototransduction activation, a desensitization mechanism is triggered to allow photoreceptors to return to basal activity and become sensitive to a new stimulus [38,39,40]. Analysis of the ERG amplitudes at 200 milliseconds (ms) after the flash exposure of different intensities revealed that *Faim* KO retinas still maintain the flash-evoked response (Figure 3B), while their WT counterparts have already returned to the resting state. This suggests that *Faim* KO mice could have a defective photoreceptor deactivation after stimulation, indicating alterations in adequate phototransduction termination.

These results prompted us to study whether dark adaptation in Faim KO retinas was impaired, as this could be the cause of a delayed photoresponse deactivation. To assess this, a double-flash protocol was performed to measure the rod photoresponse recovery time at both three (Figure 4) and 18 months of age (Figure 5). 

The following full-field ERG resolved that *Faim* KO mice present a severe delay in dark-adaptation at both ages in comparison to WT mice (Figure 4A.i for three-month-old and Figure 5A for 18-month-old analyses). We detected a significant delay in mixed b-wave recordings in *Faim* KO mice at 3 months of age (Figure 4C) and a trend to a delay that rendered non-significant in the mixed a-wave recordings (Figure 4A.i,A.ii,B). At 18 months of age, both mixed a- and b-waves showed a severe delay in dark adaptation (Figure 5). These results indicate that *Faim* KO mice require more time to achieve full recovery of rod-driven responses compared to WT mice. 

We then subjected young adult mice to mild LD data to study whether light stress exacerbated this dark adaptation deficit, given the reported effects of light stress on photoreceptors light and dark adaptation. LD increased the mixed a-wave dark adaptation time selectively in *Faim* KO mice (Figure 4A.i, A.ii,B), while no effect was observed on the mixed b-wave (Figure 4A.iii,C). Statistical analysis showed that the interaction factor between genotype and LD in photoreceptor response (a-wave) was significant (Figure 4B, a-wave, Genotype x LD, *p* = 0.013*), indicating that mild light exposure accentuates the dark adaptation deficiency in *Faim* KO mice, while WT remained unaltered after light stress.

### 3.3. Light-Dependent Arrestin-1 Translocation Is Impaired in Faim KO Mice

To study the mechanism that could be lying beneath the delay in dark adaptation in *Faim* KO mice, we aimed to study the photoreceptor desensitizing process. 

Light stimulation in dark-adapted mice triggers the translocation of Arrestin-1 and Transducin-α between the inner segment (IS) and the outer segment (OS) of photoreceptors in opposite directions [41,42] (Figure 6A,B).

To assess whether this translocation is altered in young adult *Faim* KO mice, Arrestin-1 (Figure 6A) and Transducin-α (Figure 6B) immunolocalization were examined in retinal sections in dark-adapted mice or mice exposed to 2500 lux of white LED Light for 5 or 15 min after overnight dark-adaptation.

Arrestin-1 in dark-adapted mice was mainly localized to the photoreceptor synaptic terminals and to the IS in WT mice, while *Faim* KO mice also exhibited Arrestin-1 immunolocalization at the cellular body (Figure 6A). After 15 min of light exposure, 70% of Arrestin-1 was translocated to the OS in WT mice, but *Faim* KO mice roughly exhibited 20% of Arrestin-1 to the OS (Figure 6A,C). On the contrary, transducin-α translocation levels from the OS to the IS were similar between *Faim* KO and WT mice (Figure 6B,D). 

Moreover, we also checked Arrestin-1 and Transducin-α protein levels at 3 and 18 months of age and found a reduction of both Arrestin-1 and Transducin-α protein levels only at 18 months in *Faim* KO retinas (Figure 7). These results suggest that the delay in Arrestin-1 redistribution that we found at earlier ages stems from a functional deficit and not a decrease in its protein levels.

### 3.4. Both FAIM Isoforms Prevent Arrestin-1 Ubiquitination In Vitro

To perform its anti-apoptotic function, FAIM blocks XIAP autoubiquitination by binding its E3 ligase domain [10,43,44]. This affinity binding stabilizes the levels of XIAP and allows the latter’s ultimate function: caspase-3 inhibition [10]. Members of the Arrestin-1 family have also been found as a target of ubiquitination for signaling purposes [45,46], and E3 ubiquitin ligase proteins are reported to act as Transducin-α translocation modulators during light and dark adaptation [21].

Since ubiquitination has been reported as a regulator of light and dark-dependent translocation of proteins involved in phototransduction in rod photoreceptors, and FAIM regulates XIAP by blocking its ubiquitination, we tested whether FAIM could be associated to Arrestin-1 ubiquitination events. On that account, HEK293T cells were transfected with 6×His-Ubiquitin encoding plasmid together with Arrestin-1-HA alone or Arrestin-1-HA with either FAIM-L-FLAG or FAIM-S-FLAG (Figure 8). After ubiquitin immunoprecipitation and in the condition in which neither FAIM-L or FAIM-S were co-expressed, Arrestin-1-HA ubiquitination was abundant. However, FAIM-L and FAIM-S co-expression almost completely abrogated Arrestin-1-HA ubiquitination (Figure 8, left panel). The same ubiquitination assay was performed under the effects proteasome inhibitor MG132 (Figure 8, right panel), which increased the levels of ubiquitinated Arrestin-1 co-expressed with FAIM-L and FAIM-S, indicating that at least some of this ubiquitination is degradative via the proteasome.

Altogether, this result suggests that both FAIM isoforms can negatively regulate Arrestin-1 ubiquitination and therefore may play a role on the retinal phototransduction signaling.

### 3.5. Photoreceptors in Faim KO Mice Are More Sensitive to LD

In *Faim* KO mice, functional alterations were only observed at 18 months of age, and a mild LD was not enough to elicit retinal alterations in young adult mice under standard ERG protocols, although a slight increase of the dark adaptation delay was found. Retinal neurodegeneration and a decrease in its functionality are commonly associated to ageing processes. We previously described that aged retinas with FAIM deficiency show increased retinal gliosis, accumulation of ubiquitinated aggregates, breakdown of the blood-retinal barrier and photoreceptor cell death [11], so we hypothesized that *Faim* KO mice may be more susceptible to these alterations at early ages after stress induction.

There are several models that mimic degenerative retinal diseases, such as light-induced retinal damage (LIRD). These LIRD models have been used to study natural ageing processes, retinal pathologies like age-related macular disease (AMD) and retinitis pigmentosa (RP) [33,47,48,49], which are characterized by photoreceptor cell death.

To evaluate whether *Faim* deletion renders retinas more sensible to neurodegenerative insults after light damage, we exposed mice to white LED light at 10,000 Lux for 8 h and 24 h and after two weeks we analyzed the effects on retinal structure. As shown in Figure 9A,C, after 8 h of LD young adult *Faim* KO mice already exhibited accumulation of ubiquitinated proteins and retinal gliosis through an increase of GFAP immunoreactivity (Figure 9B,C). Additionally, these alterations worsened after 24 h of LD. Moreover, three-month-old *Faim* KO mice exhibited a significant number of TUNEL-positive cells after 24 h LD, but not 8 h LD, while WT mice did not show photoreceptor cell death at either 8 h or 24 h LD (Figure 10). These results indicate that lack of FAIM renders the retina more sensible to light stress, suggesting that FAIM could confer protection to photoreceptors against LD.

To address whether this higher susceptibility to LD correlates with differential gene expression, we performed bulk RNA-seq with mice exposed to LD for 24 h at 8000 lux (Figure 11). The RNAseq data showed a significant increase in the number of downregulated genes in *Faim* KO retinas in comparison to WT mice in LD conditions. These genes include *Gpx3* and *Sod3*, which closely link to an adequate oxidative stress response [50,51] (Figure 11A). We also used Gene Ontology (GO) terms for the gene-enrichment analysis [52,53] in order to identify potential associations. We found that *Faim* KO mice present a different transcriptional response to WT against LD and genes linked to visual system and its development are overrepresented (Figure 11B). 

Therefore, this supports the notion that *Faim* KO mice retinas behave differently to WT ones in the presence of LD, and that the consequences of the absence of FAIM are detrimental to the retina in response to light stress.

## 4. Discussion

In the present work, we report for the first time how the mouse retina is functionally impaired in the absence of FAIM, following our previous results in which we showed how *Faim* KO mice present histopathological alterations compatible with a neurodegenerative phenotype. This is also the first time that the lack of FAIM has been reported to cause a functional deficit in the entirety of the CNS in vivo.

After assessing the retinal function through ERG analyses, we found a rod-driven malfunction at 18 months, while cone pathways remained unaltered. This is consistent with previous single-cell RNA-seq analyses reported elsewhere, which showed that *Faim* expression was heavily enriched in rod photoreceptor cells in comparison to other neuronal cell types, including cone photoreceptors [11,54]. The observed decrease of rod-photoreceptor response could be the result of the cell death that we previously reported at this age, since loss of photoreceptors is one of the main reasons for the decrease in ERG wave amplitudes in the retina [55]. Nonetheless, the relatively low number of TUNEL-positive cells that we encountered in *Faim* KO mice may not be sufficient to explain such retinal malfunction, prompting us to study alternative mechanisms at the molecular level for such deficits.

RGC responses were also reduced in aged *Faim* KO mice, although not severely. This functional defect may be due in part to RGCs receiving a reduced visual signal because of the functional loss of rod photoreceptors. Additionally, it could also be explained by the chronic pro-inflammatory phenotype and blood-retinal barrier (BRB) breakdown that we previously found in these mice, which progressively worsened with age [11]. Similarly, the retinal neurodegeneration in a tauopathy mouse model (P301S, line PS19) [56,57] presents a progression timeline that resembles the one in *Faim* KO retinas. In the P301S model, impairment of the BRB and RGC dysfunction (which is caused by abnormal tau accumulation) is observed from early ages as a consequence of the tauopathy and increases progressively, but RGC cell death only appears in aged mice [57]. Here, we similarly report RGC functional alterations in *Faim* KO mice. Nonetheless, in our case, this dysfunction is not associated with RGC degeneration, which could be due to the milder neuropathological phenotype that we encountered [3]. Hereby, we hypothesize that the BRB leakage, gliotic phenotype and protein aggregation that we found in late ages of *Faim* KO mice could be causing RGC dysfunction but may not be enough to induce cell death.

Likewise, in glaucoma, a disease characterized by progressive damage of the optic nerve caused by RGC death, it has been reported how glial activation and neurovascular alterations are associated to RGC degeneration, and that Müller cell gliosis may exacerbate RGC apoptosis [58,59,60]. Interestingly, we also found a delay in OPs implicit time at 18 months of age, a phenotype that is associated to retinal pathological conditions such as vitamin A deficiency [61] and diabetic retinopathy [62,63,64]. 

Despite the histopathological phenotype that we observed, a mild LIRD protocol did not alter retinal cell responses in *Faim* KO mice at three months. This is likely due to endogenous neuroprotective mechanisms that are conserved at early ages in *Faim* KO mice. For instance, it has been reported how the increase of EDN2 expression, a neuroprotective molecule that stimulates photoreceptor survival, is an early response elicited by LD [65,66,67]. Correspondingly, we previously reported how EDN2 expression is altered in *Faim* KO photoreceptor segments at 12, but not two–three months of age [11]. Thus, the EDN2 neuroprotective system could be intact in young adult mice, but not at later ages, compromising *Faim* KO retinas and rendering them more susceptible to stress. Moreover, the effect of the substrain in which *Faim* KO mice were bred must also be taken into account, as C57BL/6J mice are notoriously known in the ophthalmologic field for their light damage resistance [4,5]. Hence, we cannot rule out that the specific conditions of light intensity and duration of light exposure may not have been enough to induce retinal malfunction in our model.

The use of LIRD protocols and mouse models allows the replication of the damage caused by environmental exposure to high light levels, which accelerates loss of vision in retinal diseases such as AMD or inherited retinal dystrophies [68,69]. Constant light exposure causes a persistent bleaching of rhodopsin, which increases its rate of deactivation and regeneration. This leads to the deleterious production of phototoxic products [69,70] that accumulate in photoreceptors, which are particularly sensible to this kind of stress, ultimately causing their death [71]. In view of this, it could be interesting to explore whether *Faim* KO retinas present oxidative stress, both after LIRD and in normal conditions. This would provide us great insight on the possible mechanisms by which the absence of FAIM leads to photoreceptor cell death. In agreement with this, it has been reported how FAIM-deficient cell lines have an increased sensitivity to oxidative stress-induced cell death [12]. Additionally, in RNA-seq data, we observed the downregulation of genes involved in protection against oxidative stress in *Faim* KO mice after LD, such as *Gpx3* and *Sod3* [72,73]. Although these results will require further validation and additional experiments, they show promising perspectives about the relationship with FAIM and oxidative stress.

To avoid the phototoxic effects of a persistent rhodopsin signaling, it is essential to maintain an adequate photoresponse termination, which is also a key process for dark adaptation [74]. To this end, transgenic mice that present functional alterations in proteins involved in rhodopsin deactivation such as Arrestin-1 have been studied [75,76,77,78]. For instance, in mouse models in which the SAG gene (which encodes Arrestin-1) is mutated or Arrestin-1 function is compromised, photoresponse is not properly inhibited and the retina is incapable of adequate dark adaptation, which can lead to night blindness [79]. Excitingly, when we studied retinal function in *Faim* KO mice, we found a severe delay in dark adaptation at young adult and old ages, implying that FAIM could be involved in the mechanism behind dark adaptation. 

Dark adaptation allows the retina to increase its sensitivity in the dark after bright light exposure [80,81,82,83]. Following its disruption, the temporal resolution of vision is impaired, and it can lead to suboptimal eyesight and night blindness. This symptomatology is also characteristic of vitamin A deficiency [84], congenital stationary night blindness (CSNB) [85], Oguchi disease [86,87] or RP, diseases that are directly linked to the malfunction of Arrestin-1 and GRK1 [75,78,86,88], and Rhodopsin and Transducin-α [85,89,90]. These types of retinopathologies are rare, and there is no cure or treatment to date [85,88,91], which only makes the study of the molecular mechanisms behind these alterations more relevant to the ophthalmology field. 

When we assessed Arrestin-1 translocation upon light reception, which is a key event in phototransduction signaling [20,92], we observed that it was impaired in *Faim* KO mice. At the functional level, this deficit would compromise adequate rod-photoresponse termination, delaying its repolarization and photosensitivity recovery [20,92], which can be observed through the increased implicit times and longer rod responses in *Faim* KO mice. Additionally, this impaired photoresponse recovery can translate into the dark adaptation delay that *Faim* KO mice present at young and old ages. Altogether, these results suggest that FAIM is important for adequate dark adaptation through the modulation of Arrestin-1 translocation.

The mechanism underlying the kinetics of Arrestin-1 translocation is unknown, and different hypotheses have been enunciated [41,42,92,93,94]. Protein–protein interactions in phototransduction are under a strict control for adaptational responses to light exposure and photoreceptor cell viability [16], and it has been described that ubiquitin holds a role in protein trafficking. For instance, a ubiquitin-dependent regulation of Transducin-α translocation has been proposed [21]. Although we did not find alterations in Transducin-α translocation in *Faim* KO mice, its protein levels were reduced at 18 months of age. A significant reduction in Transducin-α could hinder phototransduction and even cause a decrease in mixed a-wave amplitudes, which we observed in our *Faim* KO at the same age. 

Upon seeing the roles of ubiquitin in protein translocation, previous results linking FAIM to ubiquitination [10,11,12] and after finding a specific delay in Arrestin-1 translocation in *Faim* KO retinas, we studied Arrestin-1 ubiquitination in vitro. On this basis, we described that (1) Arrestin-1 is susceptible to ubiquitination, and (2) its ubiquitination is almost abrogated in the presence of FAIM-S or FAIM-L. We also found that at least a fraction of Arrestin-1 ubiquitination blocked by FAIM is degradative. Agreeing with these results, we found a decrease in Arrestin-1 protein levels in aged *Faim* KO mice, which could mean that Arrestin-1 is excessively being targeted for degradation via the UPS in the absence of FAIM. To assert this, we could perform additional ubiquitin assays with mutated lysines to discern the type of ubiquitination that FAIM is blocking, given that ubiquitination in different lysine residues leads to different downstream signaling [95,96].

Interestingly, it has been reported that oxidative stress causes the aggregation of Rhodopsin and Transducin-α, and how this aggregation targets them for ubiquitin-mediated degradation [97,98,99]. This topic has not been extensively studied, and to the best of our knowledge, this is the first time that a visual Arrestin-1 has been shown to be susceptible to ubiquitination. Consistent with this, Rothstein and colleagues described that in the absence of FAIM, ubiquitin aggregates accumulate in vitro after oxidative stress [12]. To further explore this, it would be interesting to study whether oxidative stress generated by LD causes Arrestin-1 ubiquitination in vivo and its consequent aggregation in WT mice, and whether the absence of FAIM exacerbates this mechanism. 

## 5. Conclusions

In summary, we have observed that *Faim* KO mice present retinal malfunction at late ages and a delay in dark adaptation from early ages, which could be explained by alterations in Arrestin-1 translocation upon light reception. Considering this, it is possible that FAIM could assist Arrestin-1 translocation through modulation of its ubiquitination, similarly to the process that regulates translocation in other phototransduction proteins. Therefore, in the absence of FAIM, Arrestin-1 ubiquitination would increase, hampering its translocation and causing a delay in dark adaptation. We believe that the obstruction of the role of Arrestin-1 in rhodopsin deactivation by a lack of FAIM could lead to impaired rhodopsin deactivation and consequent regeneration, leaving photoreceptors more sensitive to age-related neurodegeneration and LIRD, ultimately causing functional alterations in rod photoreceptors.

Hereby, we propose that FAIM has a role in phototransduction termination through its ability to promote Arrestin-1 translocation, and as a result, *Faim* KO retinas present a symptomatology that is typically found in night blindness disorders. In light of this, we believe that research in proteins involved in the mechanisms underlying these diseases, such as FAIM, is of utmost importance.

## Figures and Tables

**Figure 1 cells-12-00487-f001:**
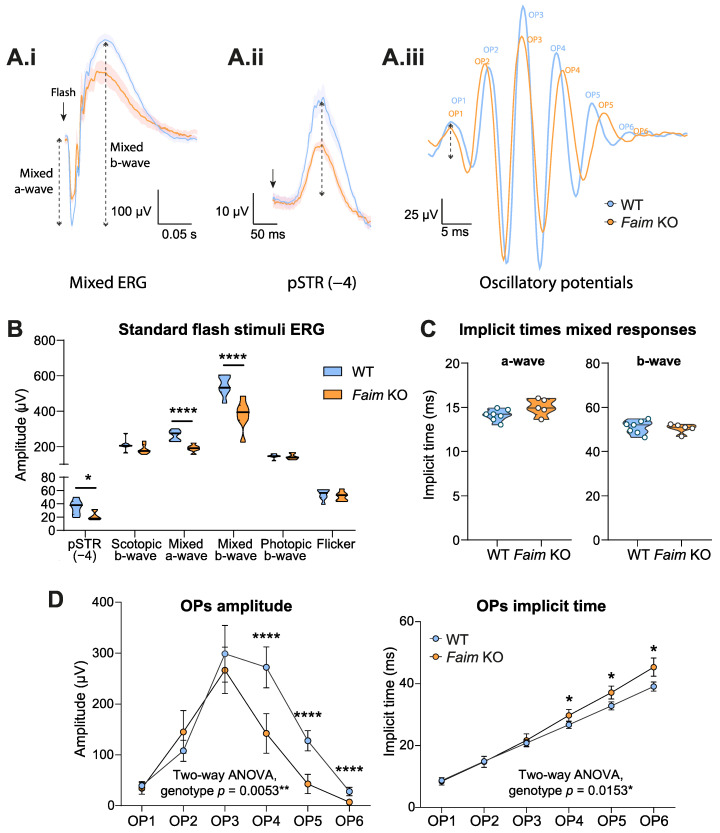
Photoreceptor and RGCs responses are altered in *Faim* knockout (KO) mice at 18 months. (**A**) Electroretinogram (ERG) recordings of mixed a- and b-wave responses at 1.5 log cd·s/m^2^ (**A.i**), positive scotopic threshold response (pSTR) (−4 log cd·s/m^2^) (**A.ii**), and representative oscillatory potentials (OPs) from each genotype (**A.iii**). (**A.i**,**A.ii**) data are plotted as the mean (solid line) and SEM deviation (shaded area) of each condition. Bold arrows indicate the stimulus flash. (**B**) Summary of ERG analysis of pSTR representing retinal ganglion cell (RGC) response, scotopic b-wave representing rod bipolar response, mixed a- and b-waves mainly representing rod responses and photopic b-wave and flicker (20 Hz) representing cone responses. To summarize, data were plotted together in B as violin plots. (**C**) Implicit times of the mixed a- and b-waves graphed as violin plots. Student’s *t* test was performed. (**D**) Summary of the amplitudes and implicit times recorded on ERG of the OPs. OP4-6 are considered to be linked to amacrine and RGC responses. Data in B and C are plotted as violin plots, in which the median is represented by a thick line. Data in D are represented as means and SD (n = 7–14 mice/group). Multiple Student’s *t* test and Student’s *t* test were performed between genotypes in B and C, and Two-way ANOVA and Sidak’s post-hoc test were performed in D. * *p* < 0.05, ** *p* < 0.01, **** *p* < 0.001.

**Figure 2 cells-12-00487-f002:**
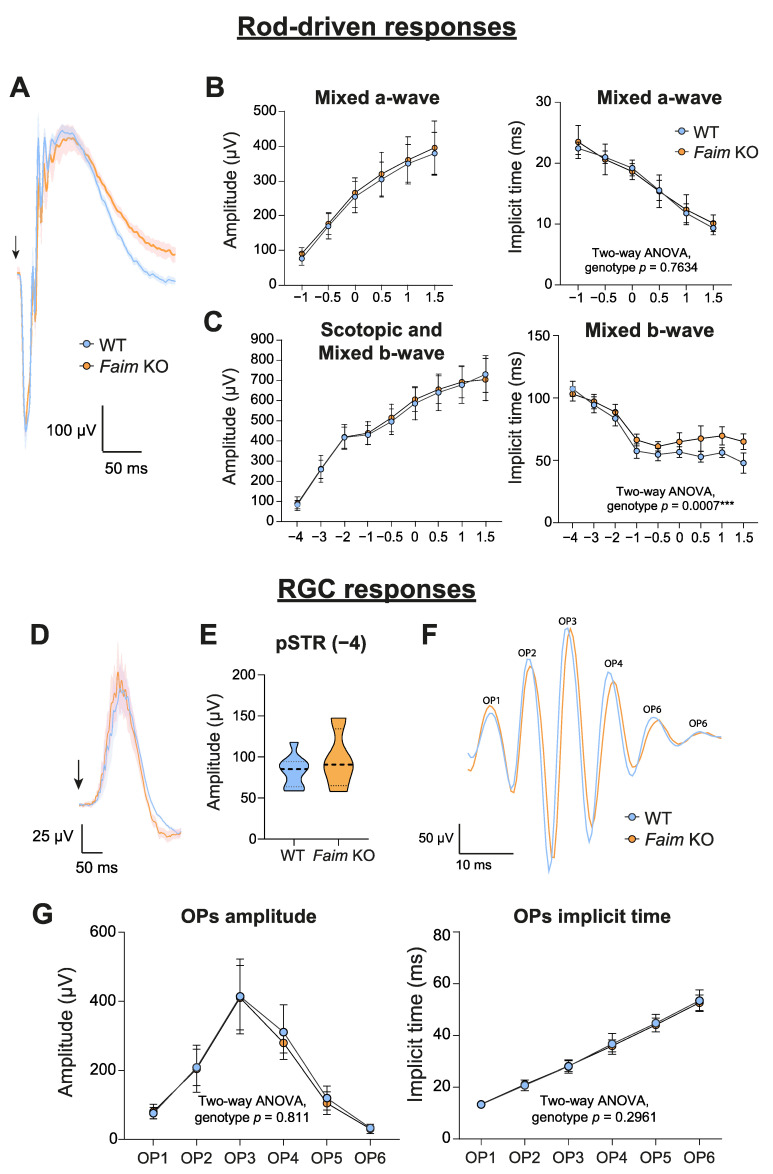
*Faim* KO mice ERG analyses are not altered at three months of age. (**A**) ERG recordings at 1.5 log cd·s/m^2^ plotted as the mean (solid line) and SEM deviation (shaded area) of each condition. (**B**) a-wave amplitude and implicit time at 1.5 log cd·s/m^2^. (**C**) b-wave amplitude and implicit time at 1.5 log cd·s/m^2^. (**D**) ERG waves of pSTR responses at −4 log cd·s/m^2^ plotted as the mean (solid line) and SEM deviation (shaded area) of each condition. (**E**) pSTR amplitudes represented in violin plots. (**F**) Representative ERG recordings of OPs at 1.5 log cd·s/m^2^ plotted as representative data of each genotype. (**G**) OPs amplitudes and implicit times at 1.5 log cd·s/m^2^. In (**B**,**C**,**G**), data are plotted as mean and SD, and in E data, are plotted as violin plots. (*n* = 8–9 mice per genotype). Bold arrows indicate the stimulus flash. Two-way ANOVA and Sidak’s post-hoc tests were performed in (**B**,**C**,**G**), and Student’s *t* test was performed in (**E**). *** *p* < 0.005.

**Figure 3 cells-12-00487-f003:**
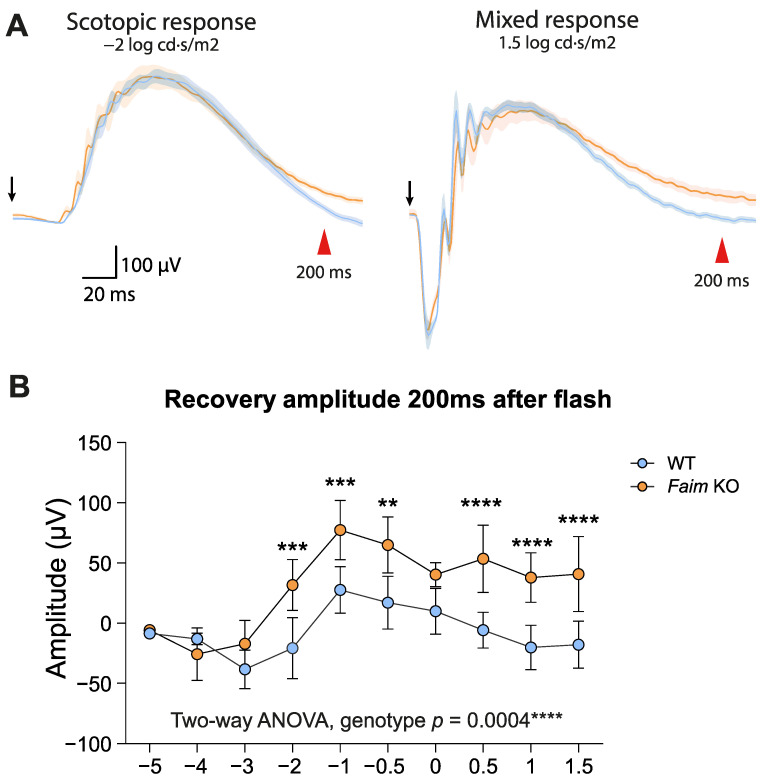
*Faim* KO mice present a delay in photoresponse deactivation at three months. (**A**) ERG recordings of scotopic and mixed responses (−2 and 1.5 log cd·s/m^2^, respectively) plotted as the mean (solid line) and SEM deviation (shaded area) of each condition. It can be observed how the recovery of the mixed response in the *Faim* KO animals is prolonged in time with respect to the WT mice. Black arrows indicate the stimulus flash. Red arrowheads indicate 200 ms timepoint. (**B**) Recovery amplitude of WT and *Faim* KO mice 200 ms after the flash was evoked. Two-way ANOVA and Sidak’s post-hoc tests were performed. ** *p* < 0.01, *** *p* < 0.005, **** *p* < 0.001 (*n* = 8–9 mice per genotype).

**Figure 4 cells-12-00487-f004:**
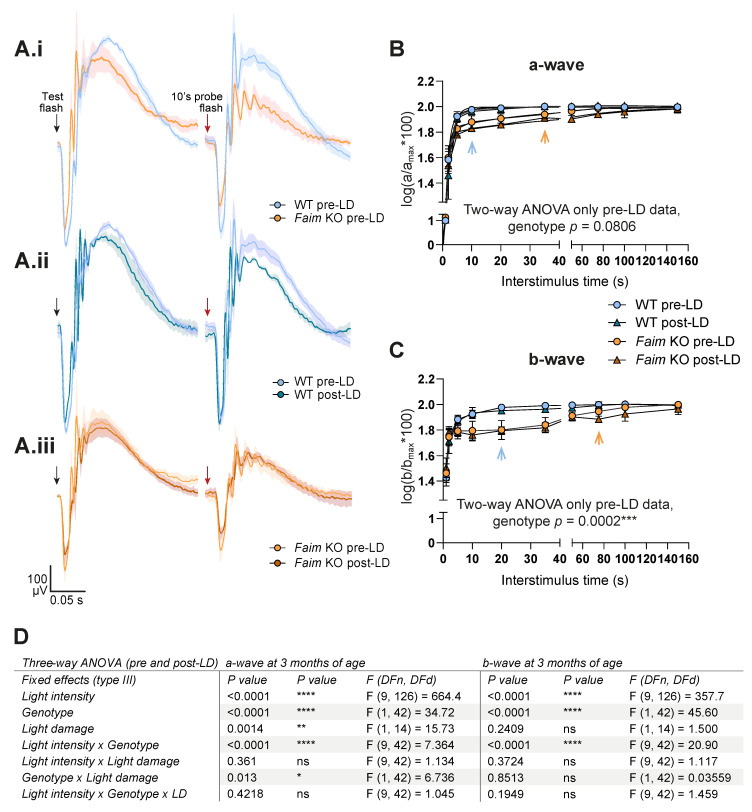
*Faim* KO mice show a great delay in dark adaptation in comparison to WT mice at 3 months, but few differences are found after LD. (**A**) ERG waves at 1.5 log cd·s/m^2^ after test flash (left, black arrow) and after 10′s probe flash (right, red arrow). Data are plotted as the mean (solid line) and SEM deviation (shaded area) of each condition. (**A.i**) ERG recordings from WT and *Faim* KO mice before light damage. (**A.ii**) ERG recordings from WT mice before and after LD. (**A.iii**) ERG recordings from *Faim* KO mice before and after LD. (**B**,**C**) Graph representing the delay in dark adaptation found in the mixed a-wave (**B**) and b-wave (**C**) of *Faim* KO mice in comparison to WT mice. Data are represented as mean and SD and as log(a/a_max_*100) or log(b/b_max_*100) to interstimulus time, where “a_max_” and “b_max_” are the a- and b-wave after the test flash, and “a” and “b” are the a- and b-wave after the probe flash. Statistical analysis was performed using two-way ANOVA when comparing only pre-LD data, and three-way ANOVA when comparing all data, including pre-LD and post-LD results. (**D**) Three-way ANOVA results of mixed a- and b-wave dark adaptation analyses. N = 6–7 mice for pre-LD analyses, and n = 4 for post-LD analyses. * *p* < 0.05, ** *p* < 0.01, *** *p* < 0.005, **** *p* < 0.001.

**Figure 5 cells-12-00487-f005:**
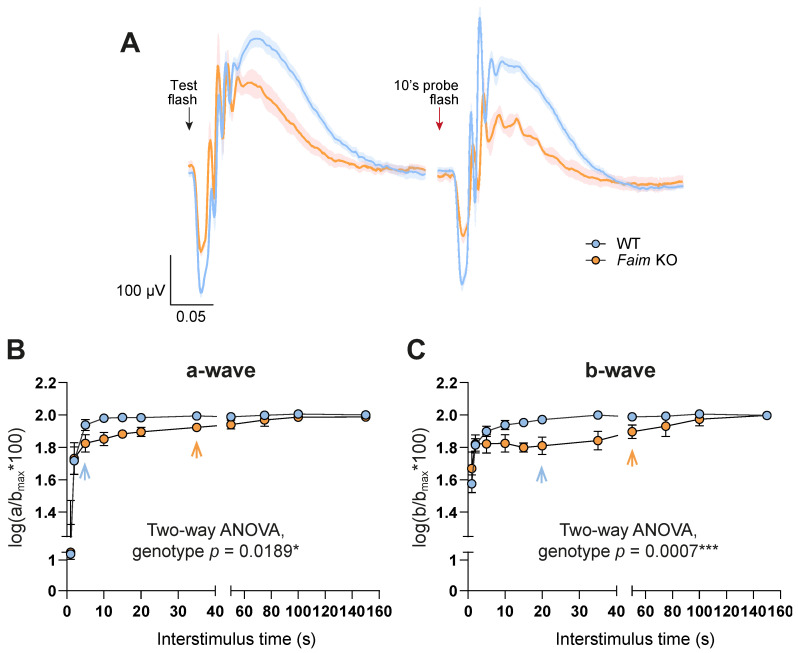
Delay in dark adaptation in *Faim* KO mice is maintained at 18 months of age. (**A**) ERG responses at 1.5 log cd·s/m^2^ after test flash and 10′s probe flash. Data are plotted as the mean (solid line) and SEM deviation (shaded area) of each condition. (**B**,**C**) Graph representing the delay in dark adaptation found in the mixed a-wave (**B**) and b-wave (**C**) of *Faim* KO mice in comparison to WT mice. Data are plotted as mean and SD and represented as log(a/a_max_*100) or log(b/b_max_*100) to interstimulus time, where “a_max_” and “b_max_” are the a- and b-wave after the test flash, and “a” and “b” are the a- and b-wave after the probe flash. Statistical analysis was performed using two-way ANOVA. *n* = 6–7 mice/group. * *p* < 0.05, *** *p* < 0.005.

**Figure 6 cells-12-00487-f006:**
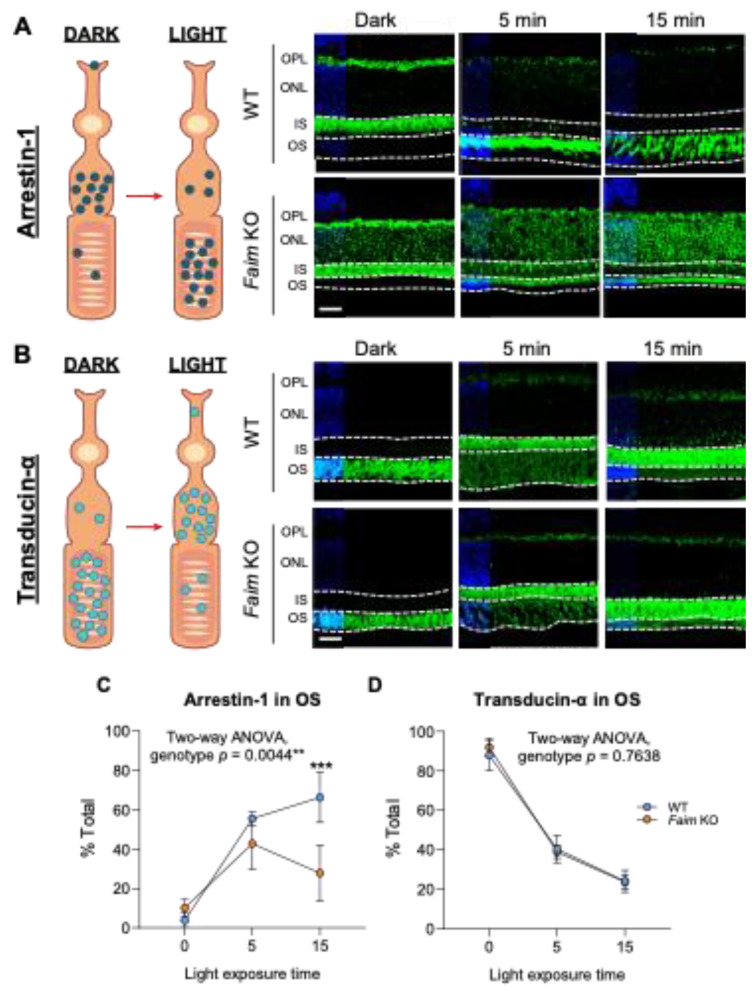
Light-dependent Arrestin-1 and Transducin-α translocation upon light reception. Diagrams representing the translocation of Arrestin-1 (**A**,**left**) and Transducin-α (**B**,**left**) from dark-adapted to light-exposed retinas. Representative images of retinal sections immunostained for Arrestin-1 (**A**,**right**) or Transducin-α (**B**,**right**) after dark-adaptation and 5 or 15 min of light stimulation. Nuclei were stained with Hoechst (blue). (**C**,**D**) Quantification of the total of Arrestin-1 and Transducin-α immunostaining in the OS relative to the total of the expression in photoreceptor segments, respectively. Arrestin-1 translocation in *Faim* KO mice was impaired in comparison to WT mice (**C**), but Transducin-α translocation remained normal (**D**). *N* = 4 mice per group. Data are represented as mean and SD and represented as total % of protein translocated in the outer segment (OS) per time of light exposure. Two-way ANOVA and Sidak’s post-hoc test was performed. *** *p* < 0.005. OPL: outer plexiform layer, ONL: outer nuclear layer; IS: inner segment; OS: outer segment. Scale bar: 20 μm. ** *p* < 0.01.

**Figure 7 cells-12-00487-f007:**
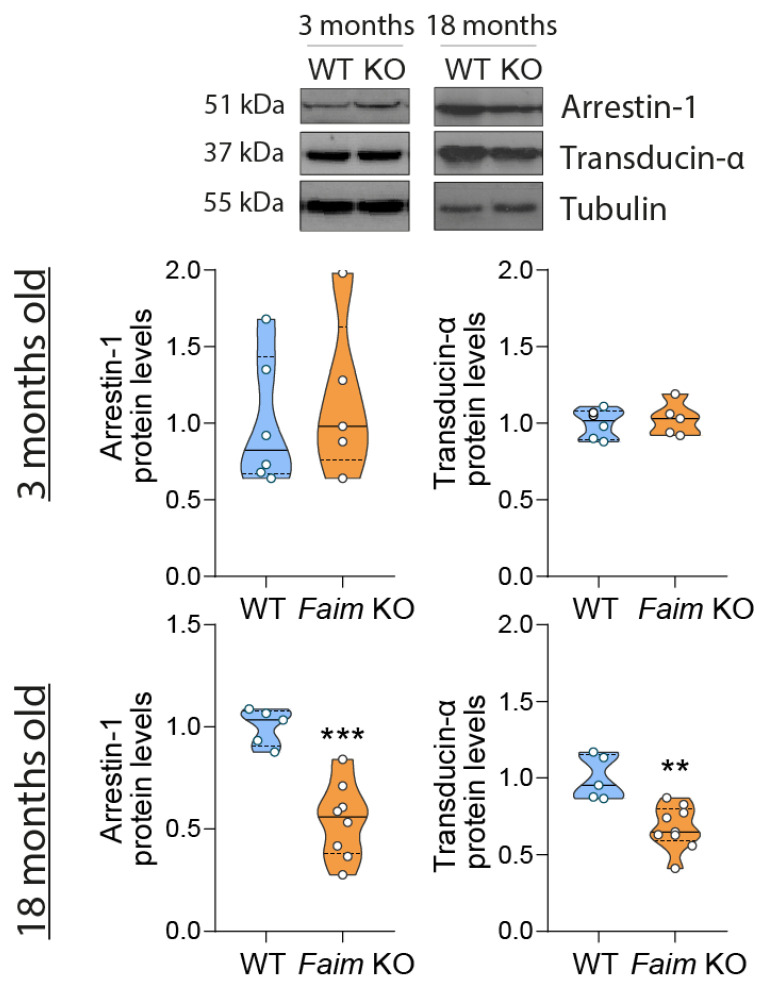
Protein levels of Arrestin-1 and Transducin-α are reduced at 18 months in *Faim* KO mice. Western blot analysis of Arrestin-1 and Transducin-α protein levels at 3 and 18 months of age in WT and *Faim* KO mice. Protein bands were quantified with ImageJ software levels and normalized to tubulin-α. Data are plotted relative to WT values and represented in violin plots. Each violin plot extends from the min to max values, the median is represented by a thick dashed line, and quartiles are represented by thin dotted lines. *N* = 5–9 mice per group. Statistical analysis was performed using Student’s *t* test. ** *p* < 0.01, *** *p* < 0.005.

**Figure 8 cells-12-00487-f008:**
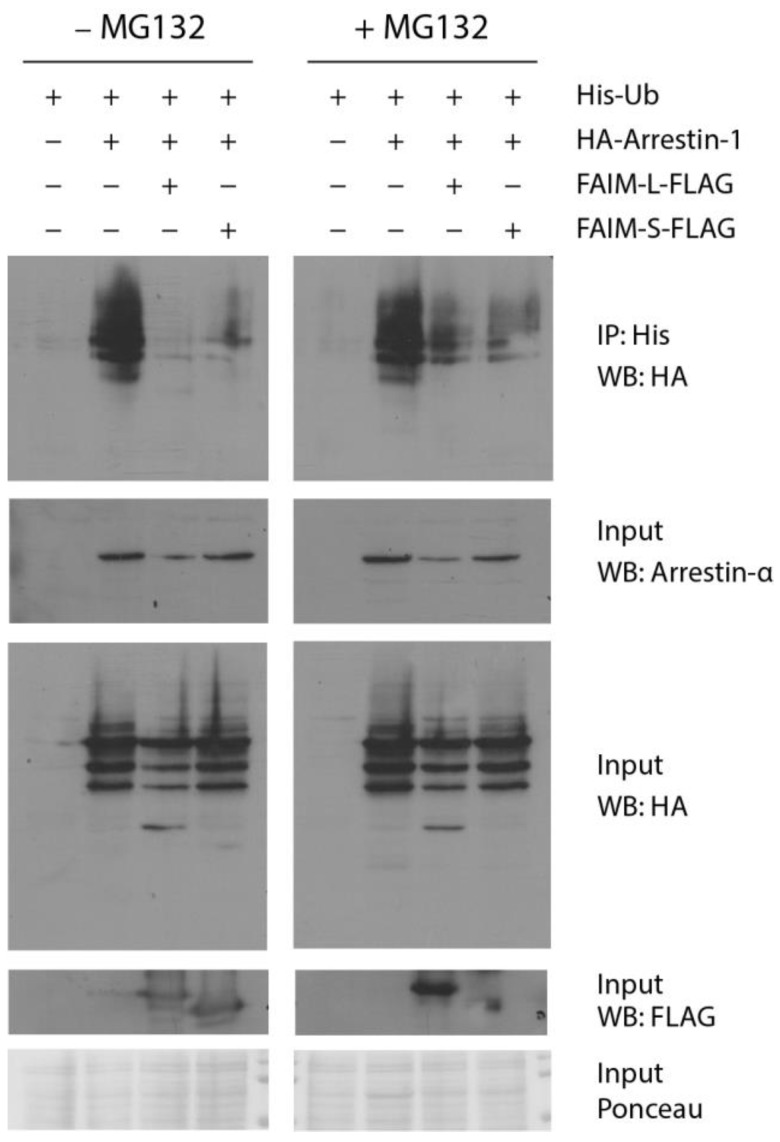
FAIM-S and FAIM-L block Arrestin-1 ubiquitination in vitro. HEK293T cells were transfected with the described plasmids, in the presence or absence of MG132 treatment, as indicated. Successful transfection was confirmed by immunoblotting for HA and FLAG for Arrestin-1 and both FAIM-L and FAIM-S expression, respectively. Ponceau staining was used to check protein loading. Ubiquitin-His pull-down were run in a gel and immunoblotted for HA. Evident ubiquitination is observed in the condition were Arrestin-1 is expressed alone with ubiquitin, and a significant decrease of ubiquitin is found in both conditions in which FAIM-L and FAIM-S are overexpressed. Data are representative of three independent experiments.

**Figure 9 cells-12-00487-f009:**
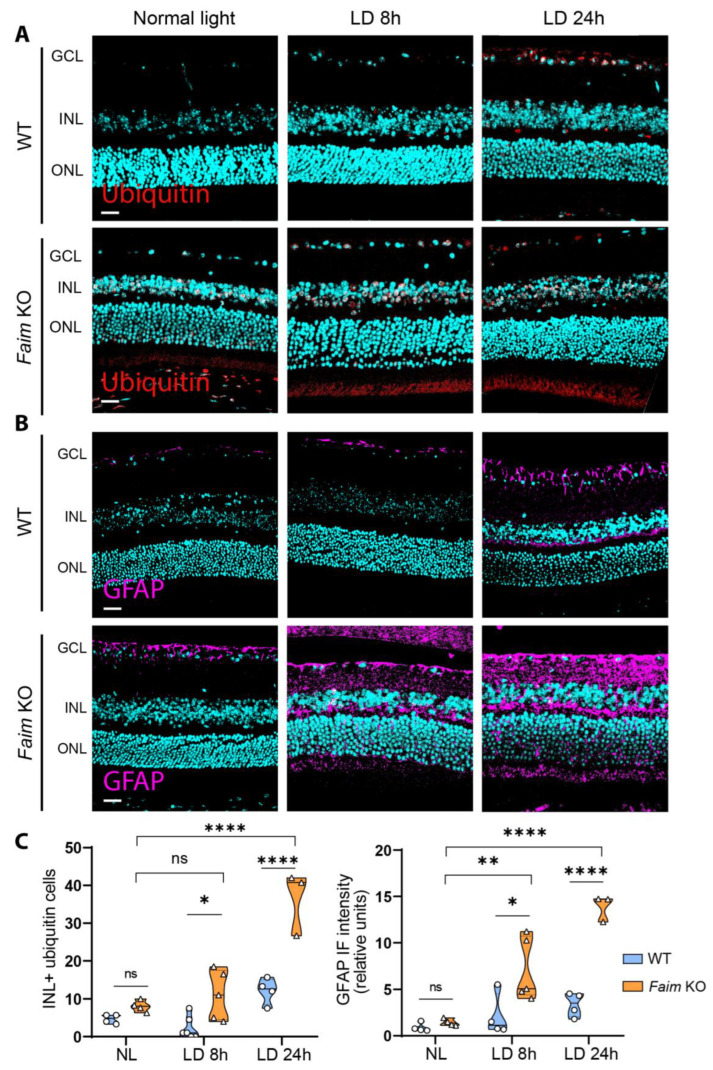
*Faim* KO retinas are more prone to accumulate ubiquitin aggregates and express more GFAP than WT mice after light damage. Representative images of 4 μm paraffin sections of retinas taken from three-month-old mice exposed to normal light and to light damage for 8 h (LD 8 h) or 24 h (LD 24 h) and immunolabeled for (**A**) ubiquitin (red) or (**B**) GFAP (magenta). Nuclei were stained with Hoechst (cyan). (**C**) Ubiquitin-positive cells in the INL were counted in at least five images per mice (left); total GFAP intensity was quantified using ImageJ software in at least five images per mice (right). Data are plotted as violin plots. Each plot extends from the min to max values and the median is represented by a thick dashed line (*n* = 4–6 mice/group). Statistical analysis was performed using Two-way ANOVA and Sidak’s post-hoc test; * *p* < 0.05, ** *p* < 0.01, **** *p* < 0.001. GCL: ganglion cell layer; INL: inner nuclear layer; ONL: outer nuclear layer. Scale bar: 20 μm.

**Figure 10 cells-12-00487-f010:**
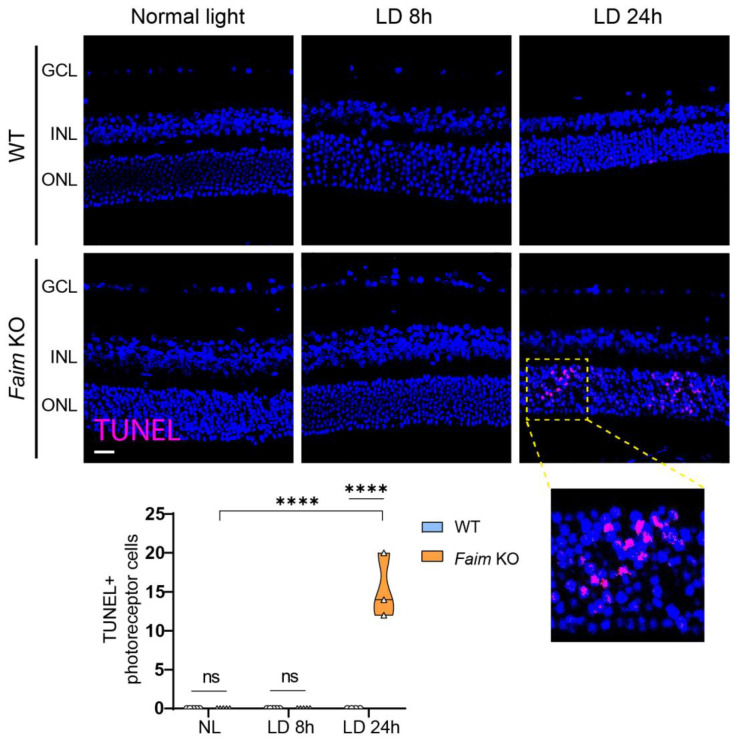
Light-induced retinal cell death in *Faim* KO mice after 24 h of light exposure at 8000 lux. Representative images of 4 μm paraffin sections of retinas taken from three-month-old mice exposed to normal light and to light damage for 8 h (LD 8 h) or 24 h (LD 24 h) and TUNEL assay was performed (magenta). Nuclei were stained with Hoechst (blue). Total number of TUNEL+ cells per section were counted. Each violin plot extends from the min to max values and the median is represented by a thick dashed line (*n* = 4–6 mice/group). Statistical analysis was performed using Two-way ANOVA and Sidak’s post-hoc test. **** *p* < 0.001. GCL: ganglion cell layer; INL: inner nuclear layer; ONL: outer nuclear layer. Scale bar: 20 μm.

**Figure 11 cells-12-00487-f011:**
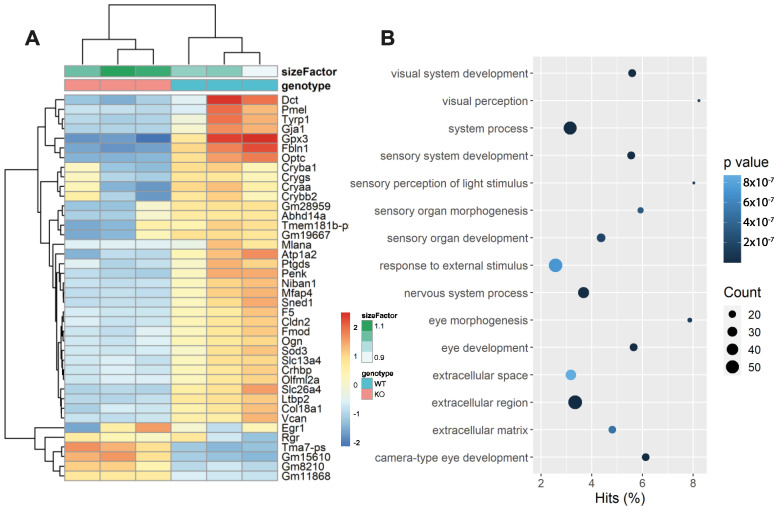
RNA-seq data analysis of WT and *Faim* KO retinas exposed to 24 h LD. (**A**) Heatmap of RNA-sequencing expression data showing the top genes that are differentially regulated following light damage (LD, 10,000 lux for 24 h) in *Faim* KO (KO) and WT mice at two–three months of age. (**B**) Top 15 GO terms enriched in *Faim* KO mice. Dot size represents the number of genes in the GO term, and color represents the *p* value.

## Data Availability

The data presented in this study are available on request from the corresponding author. The data are not publicly available due to analyses that are still ongoing.

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
