# Peer review of "The Absence of FAIM Leads to a Delay in Dark Adaptation and Hampers Arrestin-1 Translocation upon Light Reception in the Retina"

_cells, 2023, doi:10.3390/cells12030487_

Round 1

Reviewer 1 Report

It is interesting research, but they need to further supplement more results to strengthen the connections between the evidence chain. 

1. In the result section 3.1. Loss of FAIM leads to age-related rod photoreceptor and RGC functional defect. The authors should first supplement the evidence that illustrates the expression pattern of Faim in mice retina with retinal slice, including WT and KO mice. Second, they also need to demonstrate that Faim was indeed knocked out in the KO mice retina with western blot and immunostaining of retinal slice. Third, the death or alternations of photoreceptors caused by Faim KO should be illustrated. If the result of ERG indicated that RGC layer of retina was influenced by Faim KO, the histological staining of RGC should be provided.

2. In fig.7, the bands difference between two groups are not very significant, please provide more typical result of WB.

3. In the result section 3.3. Light-dependent Arrestin-1 translocation is impaired in Faim KO mice. Please provide more persuasive evidence to demonstrate that Arrestin-1 translocation was impaired by Faim KO, but not simply evidenced by retinal staining.  

4. In the result section 3.5. please provide more convincing evidence to show that the activation of GFAP after light damage, but not simply evidenced by retinal staining. Similarly, please provide more convincing evidence to illustrate the apoptosis of photoreceptors by TUNEL, such as lower magnification of retinal images.  

5. Please replace all the weird statistical diagrams with routine ones.

Reviewer 2 Report

Manuscript title: The absence of FAIM leads to a delay in dark adaptation and 1

hampers Arrestin-1 translocation upon light reception in the 2 retina

The Reviewer recommend above mentioned paper for publication in Cells after minor Revision

In general this paper is well written and the topic is of considerable interest. However, there are some suggestions that need to be addressed before publication.

This paper describes

Introduction

This section is logically written and gives a good background into the field of the study. One sentences that needs clarification is  Page 2 lines 49-51 “Although retinal degeneration has been widely studied, the molecular mechanisms underlying functional alterations and neurodegenerative events are relatively unexplored “. This sentences is too general and needs to be supported by appropriate references. There are many different types of retinal degeneration, it would be good be more specific here and maybe mention about those that are related to this study.

Methods

How many lx of red light were used in dark-adapted mice?

Page 5 lines 217-218 If Author’s did not dehydrated the samples at the first place, why did they rehydrated them later for immunofluorescence  (line 224). This needs to be clarified, because is inconsistent.

The TUNEL assay section can be shortened at least in part if the most of the staining procedure was according to manufacturer recommendation.

There is no clear information in the text about the number of the animals/eyes used for different assays (I assume that ERG was done in all mice, how about TUNEL assay and immunocytochemistry?)

Results

There is an error in line 386

Author’s should decide if they want to discuss their results in Result section or in the Discussion section (lines 443-444 and 524, 601). I like this approach, especially that this study is quite complicated, but this is for Editor decision.

Figure 9 presents apoptotic cells in retina after miced were exposed for 8h and 24h to cool LED light. It is strange that in group of mice exposed to white LED light the number of positive nuclei was the same as in WT group. Also, the image showing apoptotic nuclei in group of mice exposed to LED light for 24h is not convincing comparing to the Figure 9 graph.

Are those mice were albino?

If Author’s decide to present protein/mRNA assays along with  morphological/immunofluorescence studies they also should preformed special staining for RGC detection (for example Brn3a). If they don’t want go into those details they should focus strictly on their main goal of the study (presented at the end of introduction).

Conclusion

This is not the place for references citation.

.

Round 2

Reviewer 1 Report

The authors response all raised issues properly, so my suggestion for the current manuscript is accepted.